# Fashion between Inspiration and Appropriation

**Barbara Pozzo**

The Faculty of Law, Department of Law, Economics and Cultures, University of Insubria, Via S. Abbondio 12, 22100 Como, Italy; barbara.pozzo@uninsubria.it

**Abstract:** Fashion is considered an element of "cultural identity". At the same time, it has always been a dynamic phenomenon in which different styles, designs and models converged, acting both as a source of attraction for designers as well as a source of inspiration to draw and depart from in an attempt at innovation. Influences were reciprocal, with the phenomenon of *Orientalism* going hand in hand with that of *Occidentalism*. Today's discussion focuses on the vindication by various ethnic groups of ways to protect their own folklore as expression of their own cultural identity. The questions that arise are manifold. This contribution aims at framing the problem in the nowadays fashion industry as well as investigating the various possibilities of protecting folklore while preserving cultural identity. The discussion will deal with recent studies that have analyzed the various aspects of cultural appropriation. Intellectual property will be taken into consideration as a way to protect folklore. Nevertheless, this article suggests that other options for achieving protection of cultural heritage and folklore emerge in the field of *Private Governance* and *Corporate Social Responsibility* that will offer new opportunities to tackle the problem of cultural appropriation in the fashion world.

**Keywords:** cultural appropriation; intellectual property; traditional knowledge; traditional designs; private governance; Corporate Social Responsibility; folklore

---

## 1. Introduction

The topic of cultural appropriation in fashion has received much attention in recent years by media as well as by the specialist literature. This paper aims at inquiring the phenomenon of cultural appropriation in the fashion field, taking into account a historical perspective, in which famous examples are analyzed in order to understand that this phenomenon is not new and how it has developed (Part 2), and a comparative law perspective, in which different solutions are taken into consideration.

It then discusses the main features of the phenomenon in nowadays' society, illustrating those cases that have attracted the attention of the vast public (Part 3), describing the tension between the idea of folklore as an expression of cultural identity and the idea of folklore as a source of inspiration. Particular attention is devoted to those cases where *inspiration* can be understood as the creative re-interpretation by designers (Section 3.1), distinguishing them from those other cases in which cultural *appropriation* may become offensive (Section 3.2) or might lead to an undesired commodification (Section 3.3).

Part 4 aims to describe the regulatory framework of folklore at national level (Section 4.1) as well as at international level (Section 4.2) and at analyzing if this framework might be considered to be offering efficient tools to prevent cultural appropriation.

The last part of this paper (Part 5) is devoted to the search of new solutions to protect folklore in the fashion field, taking into consideration the actual needs behind the protection of folklore. It concludes that an important role may be played in the future by private governance tools and, in particular, by Corporate Social Responsibility standards.

## 2. A Look into the Past

Among the famous quotations by Coco Chanel, one stands out: «*Fashion is made to become unfashionable*» (*Life Magazine*, 19 August 1957).

Clothes, hats, shoes, and accessorizes as "fashion" expressions involve an element of volatility, of transience. At the same time, they may constitute the "identity" element of a certain culture.

Before examining the debate on "fashion" as "identity" expression, some caveats are necessary.

First, attitudes to "fashion" have changed radically in the twenty-first century in the light of globalization, technological innovation and the growth of the internet. Clothes have been increasingly approached as a means of self-expression, rather than as a signifier of status or profession (Geczy and Karaminas 2018). As a result, the identity issue in fashion is changing at different speeds in various parts of the world, depending on how much the region is connected with the global world, or instead is still anchored into its local traditions and heritage.

Secondly, the debate around the concept of "identity" itself tends to take on new perspectives. At the base of every identity aspect, such as religion, nationality, class, race, culture, gender, we believe that there is a common thread that keeps all its actors together. But in most cases, this is not true in today's society. As Kwame Anthony Appiah has highlighted in a recent book: "*Much of our contemporary thinking about identity is shaped by pictures that are in various ways unhelpful or just plain wrong*" (Appiah 2018, p. XIII). In this respect, Appiah points out that "*we are living with the legacies of ways of thinking that took their modern shape in the nineteenth century, and that it is high time to subject them to the best thinking of the twenty first*" (Appiah 2018, p. XIII).

Thirdly, the differentiation between *East* and *West*, between *Occident* and *Orient*, often referred to in this paper, requires some explanations. Nowadays, East and West are characterized by different approaches in many different areas: religion, society, and—of course—fashion. This differentiation is nevertheless fluid and not static, as it relies on ideas that have changed throughout history and that have given rise to different traditions of thought according to the perspective taken as the privileged one. Although "*East*" and "*West*" are often understood as indicating two abstract categories, as symbolic representations of two different and often opposing concepts of life and thought, of two well distinct and sometimes antithetical *Weltanschauungen*, these two geographical entities support and reflect each other. The Orient is not only adjacent to Europe, as it has been the source of its civilizations and languages. Furthermore, the Orient has helped to define Europe as its contrasting image, idea, personality, experience (Said 1978, p. 9).

That said, it is also true that in a historical perspective, through changes of "fashions", it becomes possible to grasp the social transformations that have characterized Western society (Vigarello 2017). This was true for detecting the division of society into a hierarchy (Riello 2012; Muzzarelli 1999), the rise of new social classes (Muzzarelli and Campanini 2003), the emancipation of women (Marchetti 1995), or the impact of a social crisis (Summers 2016).

On the contrary, the immutability of customs and styles of clothing is said to express the stability and political and social immobility order of the Orient (Braudel 1979). In 1926, a retired French missionary doctor described his impressions on Chinese clothing in the following way: "*With Chinese clothing, it is the same as with houses: there is no change over periods of thousands of years. In China, one ignores fashion; even the most progressive mandarins, the most elegant taitai (grandes dames), dress themselves now as did the contemporaries of Confucius; and their clothing does not differ from that of workers except in the richness of its material*" (Legendre 1926, p. 86).

This view of the East often corresponds to a stereotypical view (Steele and Major 1999, p. 1). The belief in the immutability of China seems to be a *cliché* that has continued until recently to influence Western ideas, ignoring the complex interaction and the process of westernization of clothes that over time has occurred in various forms in China.

Despite the acclaimed immutability of the costumes, influences coming from the West into the East are equally demonstrable, even if the Eastern immobility mentioned by Braudel has probably meant that Western fashion did not have the same impact on Eastern society.

The constant desire to innovate with original and effective solutions led Western designers to incorporate other cultures' distinctive looks, reinterpreted by the designer's creativity and sensitivity to other cultures. This cannot be considered a novelty, as influences from far away cultures were present in European fashion since the opening of the silk trade, dating back to the fourth century (Geczy 2013, p. 17). Fabrics, together with spices, are among the products of the first routes of commerce (Segre Reinach 2006).

This appears particularly evident, if we think that almost every word for pre-synthetic textiles derives from Middle East or Asiatic roots: "cotton" derives from the Arabic *kutun*; "taffeta" evolved from the Old French *taffetas* or medieval Latin *taffeta*, based on the Persian *tāftan* (to shine); "chintz" indicates that the word derived from the Hindi "chitta" or "chint", which means variegated or spotted. Other textiles derive their names from particular areas of the Orient: "satin" goes back to the Arabic *zaytūnī*, meaning "of Tsinhiang", a town in China; "damask" comes from *Damascus*, "angora" from *Ankara*, "cashmere" from *Kashmir*; "calico", a plain-woven textile made from unbleached cotton, derives its name from the town *Calicut*, situated in the State of Kerala in Southern India (Geczy 2013, p. 2; Nassu 2016).

It can be said that textiles began to travel between countries even before men, almost in their place and since ancient times. On the other side, beyond the exchanges and hybridizations, which constitute the essence of the textile product, there are specific traditions that closely reflect the culture in which they were born and for this very reason, textiles have always been a perfect vehicle to establish, express and maintain people's cultural identity (Segre Reinach 2006, p. 54).

Influences, sometimes reciprocal, spread out, involving not only the trade of goods and materials, but also of styles, shapes and beauty ideals. "Design appropriation" can be documented in the past, already many centuries ago (Lunde 2018).

A recent study made on the so-called 'roundel-and-pearl' motif attributes its origins to several cultures and locations in Central Asia and China between the 6th and 13th centuries (Lunde 2018). The long journey of this motif can ultimately be traced back to Egypt and the Eastern lands of the Mediterranean, in representations dated to the 1st and 2nd century. The roundel design was initially linked with the status of power and nobility, and continued to be used indicating such a prerogative in its various transplants (Lunde 2018). For its specific history, it has been claimed that the iconic roundel motif may therefore be considered to be a design with no single or pure cultural heritage claiming its provenance: "*The recurring uses of the same type of motif renders this design multicultural rather than a hybrid. The roundel motif could also be considered as belonging to an international design repertoire representing a geographical region stretching over many consumers or cultural spaces*" (Lunde 2018, p. 4).

In all the discussions that will accompany us on our journey, we must therefore remember that textiles and designs are not easy to culture-classify with certainty when considering their movements and appropriations. As it has been underlined: "*One of the most compounding issues of culture-classification is that textile designs rarely contain any concrete or definitive indication of their origin*" (Lunde 2018, p. 4).

Even if the examples could be multiplied over the centuries (Chiara 2016), coming to more modern times, the West's fascination with and assimilation of the ideas and styles of the East lived a period of great vivacity at the beginning of the XXth century, with the expansion of Colonialism (Said 1978).

In this perspective, *Orientalism* was not reproducing any objective knowledge on usages and customs of the East. It was, on the contrary, a fabrication of the West: "*The perilous voyages to Cathay and Edo, and even the narrower crossing to the mysterious harems and itinerant lifestyles of North Africa and the Middle East, gave Europe a secular heaven-on-earth, a paradise undefiled by Western civilization*" (Martin and Koda 1994).

The fascination with the East reaped its victims also in the field of fashion. The French couturier Paul Poiret (Poiret 1930), for example, became a celebrity in the first half of the 20th century, not only for dispensing women from the obligatory corset, but also for re-interpreting *Orientalism* in fashion (Pham 2013). He launched harem pants and tunics as the new fashion, while designing several hundred costumes for a three act play by Jacques Richepin, *Le Minaret*, in 1913 (Troy 2002; Geczy 2013, p. 136).

Of course, Poiret was offering a reinterpretation, his personal vision of the East, as it was pointed out: «*"The Orient" may have appeared in Oriental Studies to be a term with a concrete referent, a real region of the world with real attributes, in practice it took on meaning only in the context of another term, "the West"*» (Carrier 1995, p. 3).

It was not only the West to have fallen in love with the exotic East. On the contrary, East and West maintained to some extent a dialectical relationship, so that also the opposite phenomenon to "Orientalism", what we might call "Occidentalism", took place.

Much has been said and written about Orientalism, but little is yet known of the inverse relationship: that complex and multifaceted phenomenon that led some Eastern arts to assimilate forms and contents of a purely western matrix (Sgubin and Landini 2018).

It happened in the field of arts, but it happened also in the field of fashion.

Japan, after having influenced European Art and Design since leaving its "Splendid Isolation", was experiencing, in the time span between the late nineteenth century and the forties of the twentieth century, a conflicting attitude, poised between the thrill of novelties coming from overseas and the reassuring attachment to tradition. Kimonos which, more than any other art form, were influenced by the change of the Japanese society, began to represent, aside from traditional motifs, colorful designs that recall European artistic currents (Sgubin and Landini 2018). The *Meisen* kimonos adopted fantasies suggested by the avant-garde movements: from the Viennese Secession to the Glasgow School, from Futurism to Cubism, from Divisionism to the abstract Expressionism of Jackson Pollock. The new production was inspired by facts of contemporary history or by technological achievements, in an exciting and very surprising kaleidoscope of colors, patterns, decoration and weaving techniques, also inspired by western textile production (Sgubin and Landini 2018).

Another interesting phenomenon to analyze concerns the influences that the British Raj had on Indian fashion (Tarlo 1996). This is due to different reasons. On the one side, the British were able to produce cheap machine-made versions of the finely textured fabrics previously worn only by the upper classes, attracting new and enthusiastic customers. On the other side, aniline dyes used in Europe introduced a whole new range of colors, which held an "exotic" appeal in India (Tarlo 1996, p. 46). Finally, for some Indians, it was the very "foreignness" of European clothes that rendered them appealing (Tarlo 1996, p. 46). Dress codes would also vary, according to men and women's preferences. Indian women would generally prefer to adopt European fabrics and accessorizes, while retaining Indian styles (Tarlo 1996, p. 46). Menswear fashion followed another path. The *Sherwani*, which is considered a traditional Indian and Pakistani menswear, was born during the British Rule and is an influence of British influence on *élite* Indian menswear (Gupta 2016). It was eventually Gandhi who inspired a return to the origin and the recreation of the Indian dress (Tarlo 1996, p. 62).

On another trajectory, anthropologists like Anne Grosfilley have recently told us how "African wax prints" are usually neither made in Africa nor designed by Africans. The most emblematic African fabric is actually an invented tradition (Hobsbawm and Ranger 1983), whose story is the result of a fabulous contamination between East and West (Grosfilley 2017). The story begins in Java that was part of the nationalised colonies of the Dutch East India Company, which came under direct administration of the Dutch government in 1800. In this part of the world, beautiful garments called *Batik* were produced, that were considered the traditional fabric of Indonesia. It is important to stress how many Indonesian Batik patterns are symbolic for Javanese people and that Batik garments play an important role in traditional ceremonies. That is why UNESCO (the United Nations Educational, Scientific and Cultural Organization) inscribed Indonesian Batik on the Representative List of the Intangible Cultural Heritage of Humanity in 2009.

In the XIXth century, the Dutch developed in this area an ambitious industrial and commercial strategy in order to launch a true *industrial Batik*. Known as *Wax print*, this textile did not manage to establish itself against the *Java Batiks*, for aesthetic and economic reasons. From an aesthetic point of view, the "veined" effect caused when the dye bled into the wax cracks was not considered attractive.

Nor was the price, since the Javanese artisans were able to increase productivity by applying the wax with stamps made from copper bends rather than the traditional bamboo stylus (Grosfilley 2017).

The failure of their industrial textile products in Indonesia, induced the Dutch to look for new markets that were to be found along the way back to Europe on the western coasts of Africa, where the Dutch used to refuel. In the Gold Coast, today's Ghana, the success obtained by "Wax", the Dutch industrial version of "Batik", reproducing patterns of Indonesian origin, was enormous. The product was immediately in great demand, even outside Ghana, spreading throughout Western and Central Africa. One of the reasons for its success is to be found in the creativity of the traders, who adapted colors and designs to the African preferences to cater to the tastes of this new market (Grosfilley 2017).

In the end, Wax, the printed cotton fabric in colorful patterns, considered the African fabric *par excellence*, has become, more than anything else, a powerful symbol of the encounter between cultures. At the same time, it has become an element of pan-African unity, although its arrival in Africa can be considered purely random (Grosfilley 2017).

We might, in conclusion, wonder what can be considered as a "traditional textile". In some regions of the world, indigenous communities have been exposed to trade routes that have been active even before the arrival of Europeans: "*Such exposure has led to a «global market» that has influenced the way in which these communities behave. Textiles (from fibres and dyes to yarns and finished cloths) have been a part of this very active exchange. What could be considered traditional now, was in fact very avant-garde at the beginning*" (Meneses Lozano 2014).

On the other side, some "traditional" textiles have changed during history their role of social markers. Let us take the example of *Adinkra* and *Kente* cloth, fabrics produced by the Asante people of Ghana. They were initially markers of Asante royal power and then of Ghanaian cultural distinction. They were originally handmade and reserved for the exclusive use of the Asante ruler. Nowadays, there are cheap mass-produced reproductions that proliferate in all Ghanaian markets (Boateng 2014).

## 3. The Discussion on Cultural Appropriation in Modern Fashion Business

Fashion, as we have seen, is often the result of lucky encounters. Taking inspirations from other cultures has always been a prevailing trend in the world of fashion.

What characterizes, then, the current debate in the fashion world today as today?

Cultural appropriation, framed as the taking of intellectual property, traditional knowledge, cultural expressions, or artifacts from someone else's culture without permission (Young and Brunk 2012, p. 4; Scafidi 2005, p. 9; Brown 2003, p. 3), is a multifaceted phenomenon that characterizes various aspects of human life. Much research in this field has been devoted to Aboriginal or Ethnic art (Brown 2003, p. 44), music (Coleman et al. 2012; Hall 1997), esoteric (Nason 1997) as well as scientific knowledge (Pullman and Arbour 2012).

The same definition of cultural appropriation may have different facets (Sharoni 2016, p. 4) that point out the attribution of a different meaning or practice outside the source community, the imbalance of power between the two cultures, or even the aim to borrow not for the intrinsic value of the item, but to caricature it (Scafidi 2001, p. 824).

Cultural appropriation is connected not only with proprietary issues, but also and furthermore with identity issues: "*What really lies behind the debate about cultural appropriation is not ownership but gatekeeping – the making of rules or an etiquette to determine how a particular cultural form may be used and by whom. What critics of cultural appropriation want to establish is that certain people have the right to determine who can use such knowledge or forms, because at the heart of criticism of cultural appropriation is the relationship between gatekeeping and identity*" (Malik 2017).

Commentators agree on one common feature of cultural appropriation, that is to say that the cultural property of some peoples is in danger: "*The very cultural heritage that gives indigenous peoples their identity, now far more than in the past, is under real or potential assault from those who would gather it up, strip away its honored meanings, convert it to a product, and sell it. Each time that happens the heritage itself dies a little, and with it its people*" (Greaves 1994).

The debate in the field of cultural appropriation in the fashion industry turns around very different topics that express worries of various kinds and nature. While the use of textiles, images, patterns from other cultures might be, in some cases, at the basis of some transcultural creativity and lead to appreciation of cultural diversity (Section 3.1), in some others, the use of cultural or religious symbols to create fashion has been considered inappropriate. This may lead to harm the community where the appropriated item finds its origin (Sharoni 2016, p. 8), as in cases in which the religious or cultural symbols of other ethnic groups are borrowed for commercial purposes, disregarding the values they express. Fashion may also become offensive when items reproduce stereotyped representations of a culture, of a race or of a gender (Section 3.2). Finally, in some other cases, the borrowing of patterns, motifs or design features leads to a violation of one's cultural heritage with negative economic consequences. We have, in fact, to remember that one of the main critiques addressed to cultural appropriation is that it often implies a lack of compensation to the source community for the use of their cultural product (Sharoni 2016, p. 2) (Section 3.3).

## 3.1. Inspiration as the Creative Re-Interpretation by Designers

If we look back to the catwalks of the last fifty years, we can often find the re-interpretation of some cultural elements by designers, attracted by this or that culture.

Just to name few emblematic examples, Yves Saint Laurent, who apparently did not like travelling and whose most beautiful journeys were imagined, dedicated his Spring–Summer 1967 collection to Africa: "*He created a series of delicate gowns from a variety of materials, including wooden beads, raffia, straw, and golden thread. At a time when industrial production predominated, it was a way for the couturier to renew with artisanal techniques*" (Musée Yves Saint Laurent 2019).

Gianfranco Ferré, the famous Italian designer who was appointed Artistic Director of Christian Dior in 1989, made the first of his many trips to India in 1973, where he had the chance to visit every part of the country and to study local craftsmanship. Ferré fell literally in love with India and many were the influences in his work coming from this country. For the Fall/Winter Season (1988/1989), for example, he took inspiration from the designs of the Indian shawls of Kashmir to reproduce them on light organza fabrics, worked by Indian artisans (Fondazione Gianfranco Ferré 2019).

In 2007, John Galliano, inspired by the Opera "Madama Butterfly" and the love affair between Mr. Pinkerton and Ciociosan, presented on the catwalk for Christian Dior Haute Couture collection models wearing the Geisha-inspired makeup. As Vogue pointed out, "*Kimonos, obis, and geisha makeup were Dior-ified, transformed into delicate translations of New Look peplum*" (Mower 2007). The *Diorification* of Japanese fashion items is the exact word that exemplifies the artistic contribution of the Western Designer, that do not limit their work to a mere reproduction of an original idea stemming from another culture.

In 2010, Jean Paul Gautier dedicated his Spring/Summer catwalk to Andean cultures, to the heritage of Incas and Mayas, reinventing *sombreros*, *mariachis* trousers and sandals cut from cowboy boots (Lorelle 2010).

Ideas flow and are reinvented, in fashion as in law as in other fields of human activity, as "*nothing comes from nowhere*" (Young and Haley 2012). All these examples, in fact, show that designers may be inspired by a foreign culture and are able to re-elaborate it in a "new fashion".

But in more recent times, we do have to acknowledge that the use and misuse of cultural and religious symbols, as well as the appropriation of particular expressions of local folklore by Western designers, ended up under the magnifying glass of numerous critics.

## 3.2. When Cultural Appropriation in Fashion becomes Offensive

A series of cases have attracted the attention of the general public, as well as of academics. BBC News (Soh 2018), as many other bloggers and commentators on internet (Khopar 2018; Varagur 2017) devoted much attention to the discussion.

These cases share the common feature that the community is harmed by uses that degrade cultural or religious items because they are displayed outside their traditional setting and for purposes that are different from those for which they were originally created (Kuruk 1999, p. 773).

In 1994, Claude Eliette, Chief executive of Chanel, had to apologize for putting a verse of the Koran across the chest of Claudia Schiffer when she modelled a new evening dress in Paris. The affair, that was irreverently dubbed '*the Satanic Breasts*' threatened Chanel's exports to the Muslim world after Hasan Basri, the head of Indonesia's *ulema*, the doctors of Muslim religion, described the use of those verses as '*an insult to our religion*' (O'Shea 2013).

In 2011, a Lisa Blue swimsuit featuring a print of Hindu goddess Lakshmi caused vibrant protests among Hindus. According to the North India Times, the label was called out by statesman Rajan Zed, who "*said that it was disturbing to see goddess Lakshmi, who was highly revered in Hinduism, on a swimwear displayed by a model at a fashion show. Lakshmi was meant to be worshipped in temples or home shrines and not for pushing swimwear in fashion shows for mercantile greed of an apparel company*" (Robinson 2011; Stancati 2011). In the aftermath of the protest, Lisa Blue released a response on Facebook, in which the Label was "*offering an apology to anyone we may have offended and advise that the image of Goddess Lakshmi will not appear on any piece of Lisa Blue swimwear for the new season, with a halt put on all production of the new range and pieces shown on the runway from last week removed*" (Moss 2011).

The debate around the possibility of considering cultural appropriation as offensive went on when, in December 2012, Karlie Kloss swept down the Victoria Secrets catwalk, wearing a native-style headdress for the annual fashion show. For many Native American tribes, headdresses are a traditional symbol of respect, worn by chiefs and warriors. Each feather placed on a headdress corresponds to an act of compassion or bravery. Protests coming from different groups pointed out that for these reasons, headdresses should not be worn to advertise lingerie (Heller 2017; Nicolas 2017). Even in this case, the label, and the model herself, apologized publicly.

In the discussion that developed after these critical cases, cultural appropriation in the fashion field was accused of being racist and discriminatory: "*cultural appropriation cannot be divorced from the prevalent issues of institutional racism and discrimination. In the context of fashion, it is still a white dominated industry benefiting from picking and choosing from different cultures to then make fashionable for white middle class buyers. Sub-cultures of class or race, nationality or religion should not just be something white people can try on as a novelty. You cannot play dress-up with the reality of other people's lives or what they consider sacred*" (Mulvaney 2013).

All these cases show how the public pays more and more attention to and becomes aware of the multiple implications of the use and misuse of cultural and religious symbols. Designers, on their part, understand very well that an increasingly attentive public will direct their choices and purchases towards those labels which are able to maintain an ethically correct approach as far as these matters are concerned. It is not only a question of brand's reputation, but also a question of losing entire segments of the market, if not the entire market.

The recent case related to the Dolce & Gabbana advertising campaign in China clearly demonstrates that Western designers should adopt a multicultural approach that respects cultural and religious diversity in all its expressions, because what can be considered of particular bad taste in one culture, can be constructed as very offensive in another context leading to disastrous results. In the specific case, the three videos realized for the campaign of Dolce & Gabbana were showing a young Asian model, wearing a red sequin D&G dress, having trouble eating Italian foods such as *pizza*, *pasta*, and *cannoli* with chopsticks. Playing on a bad *double entendre* characterized by sexual innuendo, in the video featuring *cannoli*, a male narrator asked the model "*is it too huge for you?*" (Pan 2018)

The final result was that the Italian luxury company was forced to cancel the fashion show already scheduled in Shanghai, while their products were removed from several Chinese online retailers.

The lack of a multicultural approach may also lead to a reputational harm due to perpetuation of negative stereotypes (Sharoni 2016, p. 2).

Only few months ago, Gucci was accused of racism because of a $890 black-knit women's balaclava that could be pulled up over the lower half of the wearer's face. The sweater included bright red lips with an opening for the mouth, a detail widely denounced on social media as evoking blackface imagery (Hsu and Paton 2019).

This led Gucci to apologize for the offense caused by the balaclava's design. Gucci further released a statement in which the Company declared that "*We consider diversity to be a fundamental value to be fully upheld, respected, and at the forefront of every decision we make. We are fully committed to increasing diversity throughout our organization and turning this incident into a powerful learning moment for the Gucci team and beyond*" (Palmer 2019).

Gucci removed the image of the sweater from its e-commerce site and withdrew the item from all of its physical stores.

*3.3. The Commodification of Culture in the Fashion Business*

For different reasons, other cases of cultural appropriation have come to clamor. In these cases, it was not the offensive behavior towards the symbols of another culture that attracted attention, but their commodification, which has different aspects some of which are positive, others negative. In a book that made history on the matter, Susan Scafidi pointed out, "*Although outsiders' commodification of a cultural product without the authorization of the source community may dilute or destroy the product or its identification with the source community, limited communal commodification may instead enhance the value of the product by forestalling inferior copies and providing an authentic version*" (Scafidi 2005, p. 60).

Some selected cases of the last few years will better explain the values at stake, more than any merely theoretical discussion. They show how the use of artistic designs may occur without the attribution to the group they derive from and, even if the group's will is not to hinder commercialization of those products, this practice harms the group's ability to profit from commercial sales (Paterson and Karjala 2003, p. 637). As Paul Kuruk was pointing out twenty years ago, what strikes is that "*in many cases where traditional art or knowledge is exploited, the communities derive wither no economic benefits, or if they do gain something, such benefits often pale in comparison to the huge profits made by the exploiters*" (Kuruk 1999, p. 772).

In 2007, Matthew Williamson was targeted by the Ethiopian government for appropriating pattern similar to its national dress. In particular, two of the designer's outfits shown during London Fashion Week were accused to be copies of traditional Ethiopian dresses. The Intellectual Property Office in Addis Ababa released a declaration to newspapers, underlying that the patterns used by the Western stylist "*symbolise our identity, faith and national pride. Nobody has the right to claim these designs as their own*". In an article published on *The Independent* in those days, the Author doubted the morality of borrowing a country's national costume, especially when that country is one of the poorest in the world (Hailesalassie 2007). Williamson's spokesperson told *The Independent*: "*Historically, Matthew Williamson bases his collections on the idea of a modern girl who is a global traveler. Her style is in part defined by incorporating many different cultures, traditions and customs. The spring-summer 2008 season was particularly inspired by the idea of modernizing and celebrating certain traditional African fabrics and costumes*".

The comments did not take long to arrive: Ethiopians would be delighted if their traditional garments and costumes could be positively influential globally like those of India and the Far East. In this case, however, Williamson did not only take inspiration from Ethiopian traditional clothes to come up with his own unique style: "*Designers can always be inspired from different traditional and cultural reflections. But copying them directly is a crime against the rightful owners*" (Demilew 2008).

In 2011, a new case of cultural appropriation was the basis for a lawsuit in the United States. Urban Outfitters launched a series of Navajo-themed items, including underwear jewelry and flasks with traditional patterns, much to the discontent of the Navajo Nation, that issued a *cease and desist letter* to Urban Outfitters followed by a lawsuit in 2012 (United States District Court, D. New Mexico. The Navajo Nation, et al., Plaintiffs, v. Urban Outfitters, Inc., et al., Defendants. Civ. No. 12-195 BB/LAM, Signed 19 September 2016).

The Navajo Nation has taken advantage of trademark law, registering the name "Navajo" as a trademark in 1943 (Fowler 2013). Navajos could, therefore, assert that Urban Outfitters used false advertising that implied trademark infringement as well as a violation of the federal *Indian Arts and Craft Act*. The *Indian Arts and Crafts Act* of 1990 (P.L. 101–644) is a truth-in-advertising law that prohibits misrepresentation in the marketing of Indian arts and crafts products within the United States and makes it illegal "*to offer or display for sale, or sell any art or craft product in a manner that falsely suggests it is Indian produced, an Indian product, or the product of a particular Indian or Indian Tribe or Indian arts and crafts organization*".

Eventually, the Navajo Tribe and the fashion company reached an undisclosed settlement in 2016 (Vézina 2019). The problem remains open for all those tribes that do not want to trademark their cultural property, because for various reasons "*the requirement that U.S. trademark be used in commerce may itself be offensive if the name or symbol in question is something that is sacred, secret, or otherwise not an appropriate subject for commercialization*".

In 2015, the French scene offered a new case of cultural appropriation to think about (D'Hoop 2017). The community of Santa Maria Tlahuitoltepec accused the French designer Isabel Marant of "plagiarism", after the launch of Marant's Spring/Summer 2015 collection, which featured a blouse very much resembling the traditional "*Huipil*" design of the Mexican Community. A group of *Mixe* women organized a press conference in which they denounced, "*Isabel Marant is committing a plagiarism because the Etoile spring-summer 2015 collection contains the graphical elements specific to the Tlahuitoltepec blouse, a design which has transcended borders, and is not a novel creation as is affirmed by the designer*" (Larson 2015).

The interesting aspect of the story lies in the fact that Marant admitted that the design was inspired by the work of the Mexican community and withdrew the blouse from sale as requested, but also announced to be currently sued by Antik Batik for copyright infringement.

Antik Batik, a French fashion brand founded in 1992, presents itself on its own website as a label that aims "*to embody the dandy elegance of a globetrotting woman*", and whose designer invents "*a wardrobe that helps you remember the best moments of a traveler who has visited the whole world*" (https://it.antikbatik.com/about-us.html).

In that particular situation, Antik Batik was claiming to own the copyright to the same indigenous garment that the Community of Santa Maria Tlahuitoltepec was vindicating as part of its own traditional heritage.

The question was decided by the District Court of Paris (Tribunal de Grande Instance de Paris, 3e Chambre, 4e Section, 3 décembre 2015, No. 15/03456) that ruled on the side of Isabelle Marant, holding not only that the design came from the said village, but that Antik Batik could not claim any property rights on it either.

### 3.4. Making the Point about Cultural Appropriation in Fashion

All these cases show how consumers are becoming increasingly attentive to the ethical aspects of fashion. As it has been underlined, there is a resurgence for the respect for the handmade especially in the West. Some authors have suggested that this is a search for authenticity in a mass-produced world, supported by a shift to ethical purchasing balanced with a desire for knowledge about the maker of the craft artefact (Littrell 2015, p. 458).

In a market like the fashion one, where the designer's reputation assumes a value equal to that of his creativity, attention to these issues will only increase in the coming years.

This leads us to analyze the problem of how to protect folklore as a form of traditional cultural expression. We will first take into consideration the regulatory framework of reference, in order to identify which solutions are available, whether intellectual property rights can offer a valid protection, or we should look for new solutions.

## 4. The Regulatory Framework

The discussion that took place in the Academy and that reflects the debate launched by magazines, newspapers and bloggers, needs to be contextualized in a broader scenario, in which native cultures have gained more and more recognition at national and international level.

Fashion is strongly related to this scenario, as it is often fed by the creativity of native cultures that is expressed in works of folklore. Folklore has entered the agenda of International Institutions in order to provide a better protection, in the light of the recognition that it embodies creativity and is part of the cultural identity of indigenous and local communities (Brown 2003; Young and Brunk 2012; Ziff and Rao 1997).

We are going to use the term "folklore", although its definition has uncertain boundaries and does not have universal acceptance (Asmah 2008, p. 272). The Merriam-Webster Dictionary defines it as "*traditional customs, tales, sayings, dances, or art forms preserved among a people*" (https://www.merriam-webster.com/dictionary/folklore).

Kuruk (Kuruk 1999, p. 777) summarizes the debate around the definition of folklore. According to William Bascom, for example, "*the tern folklore has come to mean myths, legends, folk tales, proverbs, riddles, verse, and a variety of other forms of artistic expression whose medium is the spoken word*" (Bascom 1959). Theodor Gastor adds that folklore "*is that part of a people's culture which is preserved, consciously or unconsciously, in beliefs and practices, customs and observances of general currency; in myths, legends, and tales of common acceptance; and in arts and crafts which express the temper and genius of a group rather than of an individual. Because it is a repository of popular traditions and an integral element of the popular "climate", folklore serves as a constant source and frame of reference for more formal literature and art; but it is distinct therefrom in that it is essentially of the people, by the people, and for the people*" (Gastor 1959). According to other Authors, folklore of physical objects includes the shapes and uses of tools, costumes and the forms of villages and houses (Taylor 1959).

All definitions acknowledge some common features of folklore: the mode of transmission through generations, either orally or by imitation, the fact that it is generally not attributable to one individual subject and that it is continually developed within the indigenous community (Kuruk 1999, p. 777; Asmah 2008, p. 272).

According to the World Intellectual Property Organization (WIPO), "expressions of folklore" are included in the definition of Traditional Cultural Expression (TCE) that may refer to music, dance, art, designs, names, signs and symbols, performances, ceremonies, architectural forms, handicrafts and narratives, or many other artistic or cultural expressions (https://www.wipo.int/tk/en/folklore/).

The aims of this accentuated attention to these themes stemmed from an awareness raising process of a set of factors.

First of all, native cultures find in folklore an essential mean of social identity. Originally, folklore was protected by customary law. As Kuruk recalls, rights in folklore are generally recognized "*under social criteria depending upon the degree of the kinship, age, sex, title or role of individuals in the society, and are enforced either by sanctions based on common interests or a system of magical or religious beliefs*" (Kuruk 1999, p. 781.) Observance to the norms that protect folklore is secured through a system of sanctions that may vary according to the degree of kinship, but which is not effective outside the group (Kuruk 1999, p. 786).

Second, there has been an increase in the commercial exploitation or appropriation of folklore or other forms of traditional cultural expressions, especially since new technologies were making folklore increasingly vulnerable to exploitation and misuse (Kuruk 1999, p. 770; WIPO 1982). This is a consequence of the different relationship between art and its mechanical reproduction, that has completely changed in our days (Benjamin 1936). As Paul Valéry was already pointing out at the beginning of the last century: "*Our fine arts were developed, their types and uses were established, in times very different from the present, by men whose power of action upon things was insignificant in comparison with ours. But the amazing growth of our techniques, the adaptability and precision they have attained, the ideas and habits they are creating, make it a certainty that profound changes are impending in the ancient craft of the*

*Beautiful. In all the arts there is a physical component which can no longer be considered or treated as it used to be, which cannot remain unaffected by our modern knowledge and power. For the last twenty years neither matter nor space nor time has been what it was from time immemorial. We must expect great innovations to transform the entire technique of the arts, thereby affecting artistic invention itself and perhaps even bringing about an amazing change in our very notion of art*". (Valéry 1934).

Third, business that are able to exploit these traditions have generally no connection with the communities that have developed the traditions in question. Furthermore, the result of this commercial exploitation does not generally improve the life of the communities who were the custodians of these traditions, but ends up in an economic benefit for persons not belonging to the communities (Kuek 2005).

Fourth, the commercialization process of these forms of traditional expressions has sometimes led to situations considered degrading and culturally or spiritually offensive for the communities involved (Kuek 2005).

Last but not least, the acknowledgement that the Western Intellectual Property law system, which finds its origins in the context of the European Enlightenment, focuses on the individual genius, but neglects the property rights of a group, rendering folklore difficult to protect according to its basic schemes (Scafidi 2005, p. 11).

### 4.1. The National Initiatives

Notwithstanding the skepticism towards IP tools, it is exactly on this ground that the first attempts were made to protect folklore in an effective way (Von Lewinski 2008).

From a historical point of view it is noteworthy to underline that the first attempts to regulate the use of creations of folklore originate in the period of decolonization and find place in the framework of national copyright laws of those States that were afraid that their folklore would be depleted (WIPO 1997).

Tunisia, after having reached independence from France in 1956, was the first country to provide protection for folklore within its copyright law in 1966, whose art. 6 stated: "*1. Folklore is part of the national heritage. 2. Except for national public legal persons, the direct or indirect fixation of this folklore with a view to its lucrative exploitation, requires an authorization from the Department in charge of Cultural Affairs, which may require for this fixing, a royalty fee under conditions to be determined by decree. 3. The total or partial transfer of the copyright in a work inspired by folklore, or the exclusive license for such work, is valid only if it has been approved by the Department of Cultural Affairs. For the purposes of this law, "work inspired by folklore" means any work composed using elements borrowed from the traditional cultural heritage of the Republic of Tunisia*" (Loi N° 66-12 du 14 février 1966 relative à la propriété littéraire et artistique, Journal Officielle de la République Tunisienne, 15 février 1966, p. 226).

Following the example of Tunisia, many developing countries in Africa expressed the need for a legal mechanism for the protection of folklore enacting provisions in their copyright laws (Matip and Koutouki 2008, Kutty 2007).

Morocco promulgated such a law in 1970 (Dahir n° 1-69-135 du 25 joumada I 1390, 29 juillet 1970 *relatif à la protection des œuvres littéraires et artistiques*, Bulletin Officiel n°3023 p. 1378), where folklore is considered part of the national heritage (art. 10.1.), that includes unpublished works whose identity is unknown but for which it is reasonable to assume that the author is or was a Moroccan resident (10.4.). The direct or indirect fixation of folklore with a view to its lucrative exploitation is subject to the prior authorization of a Committee provided for by the same law, against payment of a tax the proceeds of which will be spent for purposes of general or professional interest, under the conditions that will be specified by decree (art. 10.2.). The total or partial surrender of the right to use a work inspired by folklore, or the exclusive license for such work, is valid only if it has been approved by the Committee established by the law.

The Law of 1970 was finally replaced by a new Law in 2000 (Dahir n° 1-00-20 du 9 kaada 1420, 15 février 2000, portant promulgation de la loi n° 2-00 *relative aux droits d'auteur et droits voisins*,

Bulletin Officiel n°4810, p. 604), where art. 7 further amplifies the protection of folklore, stating, for example, that in all printed publications, and in connection with any communication to the public of an expression of identifiable folklore, the source of that expression of folklore shall be appropriately indicated by the mention of the community or geographic place whose expression of the folklore is used (art. 7.3).

In Ghana, the Law on Copyright of 1985 defines folklore as "*all literary, artistic and scientific work belonging to the cultural heritage of Ghana which were created, preserved and developed by ethnic communities of Ghana or by unidentified Ghanaian authors, and any such works designated under this Law to be works of Ghanaian folklore*" (Kuruk 1999, p. 778).

Soon after, other laws were promulgated that introduced elements of protection of folklore, not only in Africa (Zografos 2010).

All these new legislative texts consider works of folklore as part of the cultural heritage of the nation. They further have another common element: folklore must have been created by authors of unknown identity, but presumably being or having been nationals of the country.

The Document that was issued in occasion of the UNESCO-WIPO World Forum on the protection of folklore that took place in 1997 (WIPO 1997) provides a useful guide to the common features that the laws protecting folklore present.

Meanwhile, the idea that folklore needed a better protection was entering the international agenda of the 1967 Stockholm Diplomatic Conference for the Revision of the Berne Convention for the Protection of Literary and Artistic Works (The Berne Convention) (Zografos 2010). Although only in a limited way, the Stockholm Diplomatic Conference of 1967 reflected, for the first time, the aspirations of the developing world on protection of folklore when it adopted the following provisions in Article 15 (4) of the Berne Convention:

(a)　*In the case of unpublished works where the identity of the author is unknown, but where there is every ground to presume that he is a national of a country of the Union, it shall be a matter for legislation in that country to designate the competent authority which shall represent the author and shall be entitled to protect and enforce his rights in the countries of the Union.*

(b)　*Countries of the Union which make such designation under the terms of this provision shall notify the Director General of WIPO by means of a written declaration giving full information concerning the authority thus designated. The Director General shall at once communicate this declaration to all other countries of the Union.*

Anyway, the attempt to preserve folklore as community-owned cultural heritage through copyright laws failed because of a series of reasons. It is, in fact, quite difficult to protect knowledge and tradition handed down from generation to generation over a period of time and collectively owned by the community through a legislation that is based on the principle of originality, on a term of protection restricted to the life time of the author plus a limited period after his death and the concepts of "*author*" and "*work*", that are indistinguishably linked with a different legal tradition (Kutty 2007, p. 5).

Another noteworthy initiative was the Tunis *Model Law on Copyright for Developing Countries*, adopted in 1976 to provide developing countries with a text of a model law to assist them when framing or revising their national copyright legislation. The Tunis Model Law goes a few steps further than the Berne Convention by explicitly including folklore in the list of protected work (Zografos 2010). The Tunis Model Law defines folklore as "*all literary, artistic and scientific works created on national territory by authors presumed to be nationals of such countries or by ethnic communities, passed from generation to generation and constituting one of the basic elements of the traditional cultural heritage*" (§18 Tunis Model Law on Copyright for Developing Countries).

In order to overcome the difficulties encountered to protect folklore under the Berne Convention, the Tunis Model Law has introduced some exceptions, as in the case of "fixation". Fixation is a frequent requirement in *common law* countries, but it could not possibly be applied to folklore. Works of folklore form part of the cultural heritage of peoples and their very nature lies in their being handed on from

generation to generation orally and have never been recorded; the fixation requirement might, therefore, destroy the protection of folklore. For this reason, the Tunis Model Law, after having established in Subsection (5) the general principle that " … *a literary, artistic or scientific work shall not be protected unless the work has been fixed in some material form*", provides an explicit exception for folklore.

In the African context, the process of providing specific protection to folklore led to the establishment of a regional arrangement on folklore (Asmah 2008). The idea of drafting a regional arrangement on folklore originated in the fact that several African countries protect folklore under intellectual property and a general agreement would strengthen these efforts.

The situation was somewhat fragmented, because of the post-colonial period. There were, in fact, several regional organizations in Africa under whose umbrella the issue of folklore could be tackled. These included: the *Economic Community of West African States* (ECOWAS) and in the intellectual property areas the *African Regional Intellectual Property Organization* (ARIPO), formed among English-speaking Africa, and the *African Intellectual Property Organization* (OAPI), formed among French-speaking Africa by the adoption of a convention signed in Bangui in 1977 (Asmah 2008, p. 8; Kuruk 1999, p. 806).

It should not be surprising that these intellectual property groupings were not able to establish a common and uniform folklore protection, if we think that common law and civil law do not share the same attitude towards copyright law and the traces of the two Colonial Powers, France and England, had left different heritages in different parts of Africa.

If we look more in detail, ARIPO, that had been established by the Lusaka Agreement in Zambia on December 9, 1976, concerns only patent law. The two protocols that were adopted to this convention: the Protocol on Patent and Industrial Designs within the Framework of the African Regional Industrial Property did not address the protection of folklore.

The OAPI was created in Libreville (Gabon) in 1962. In 1977, the Libreville Agreement was replaced by a new convention signed in Bangui (Central African Republic), that deals not only with patents but also with trademarks and copyrights (Kuruk 1999, p. 807).

OAPI includes extensive provisions on folklore protection in its Annex VII. In this agreement, folklore is defined as works that are "*created by the national ethnic communities in member states which are passed from generation to generation*" (Kuruk 1999, p. 811). According to OAPI, works of folklore are considered part of the national heritage: As a consequence, their exploitation is conditioned on notice to the appropriate state agency. Fees collected for such exploitation are aimed at financing social and cultural activities.

In conclusion, we might try to identify the reasons for the struggle to protect folklore within IP and—at the same time—to focus on the concrete difficulties to reach effective results.

The reasons that have pushed towards an IP solution are easy to grasp. First, there is an evident similarity between intellectual property law works and folklore, which has naturally led to the attempt to include folklore in the framework of IP protection. Second, in a globalized world where IP has a well-established law system, it would have been easier to connect folklore protection to such a pre-existing and well known system.

The concrete difficulties that arose derive from the main characteristics of folklore itself that have already been underlined: the non-individual nature, the oral transmission, the difficulty in putting a time limitation to its protection that collide with the typical features of IP.

A further obstacle, which rendered difficult the applicability of national IP laws in this framework derives from the inadequacy of national regimes to cope with a globalized market that pushed towards the creation of international regimes.

### 4.2. International Initiatives

A series of initiatives at international level have reinforced the awareness of the many problems connected to the protection of folklore from different perspectives.

In particular, the World International Property Organization worked hand in hand with UNESCO to elaborate the *Model Provisions for National Laws on the Protection of Expressions of Folklore Against Illicit Exploitation and Other Prejudicial Actions*, that were released in 1982 and which signaled the beginning of a sui generis regulation around expressions of folklore (*Model Provisions for National Laws on the Protection of Expressions of Folklore Against Illicit Exploitation and Other Prejudicial Actions*, retrieved from https://www.wipo.int/edocs/lexdocs/laws/en/unesco/unesco001en.pdf).

The Model Provisions elaborated by UNESCO and WIPO acknowledge that folklore is an important cultural heritage of every nation, not only of developing countries. Nonetheless, folklore is of particular importance to developing countries, as a basis of their cultural identity and as means of self-expression of their peoples, both within their own communities and in their relationship to the world around them. The document further recognizes that the accelerating development of technology may lead to improper exploitation of the cultural heritage of poorer nations. WIPO and UNESCO express their worries concerning folklore, as it is commercialized by such means on a world-wide scale without due respect for the cultural or economic interests of the communities in which they originate and without conceding any share in the returns from such exploitations of folklore to the peoples who are the authors of their folklore (Kuruk 1999, p. 815).

According to the Model Provisions, "expressions of folklore" are defined as: *"productions consisting of characteristic elements of the traditional artistic heritage developed and maintained by a community or by individuals reflecting the traditional artistic expectations of such a community, in particular*:

(i)   *verbal expressions, such as folk tales, folk poetry and riddles;*
(ii)  *musical expressions, such as folk songs and instrumental music;*
(iii) *expressions by action, such as folk dances, plays and artistic forms or rituals; whether or not reduced to a material form; and*
(iv)  *tangible expressions, such as: (a) productions of folk art, in particular, drawings, paintings, carvings, sculptures, pottery, terracotta, mosaic, woodwork, metalwork, jewellery, basket weaving, needlework, textiles, carpets, costumes; (b) musical instruments; (c) architectural forms"*.

It is therefore important to underline that in particular textiles and needlework, often taken into consideration by fashion industries, are included in this definition.

The Model Provision foresee at Section 1 that the *Expression of Folklore* shall be protected against '*illicit exploitation*' and '*other prejudicial actions*'.

According to Section 3, the following utilizations of the expressions of folklore are subject to authorization when they are made both with gainful intent and outside their traditional or customary context:

(i)   any publication, reproduction and any distribution of copies of expressions of folklore;
(ii)  any public recitation or performance, any transmission by wireless means or by wire, and any other form of communication to the public, of expressions of folklore.

Section 4 provides an exception to this general principle in case of "fair use", which takes place if the utilization is made for the purposes of education; by way of illustration in the original work of an author, provided that the extent of such utilization is compatible with fair practice; by borrowing expressions of folklore for creating an original work.

It is further to underline that the Model provision foresees the establishment of 'competent authorities' in each country, while they do not deal with the question of the ownership of expressions of folklore, since this may be regulated in different ways from one country to another

In 1989, the UNESCO General Conference adopted a *Recommendation on the Safeguarding of Traditional Culture and Folklore to Governments* (UNESCO 1989), which provided a very broad definition of folklore: *"Folklore (or traditional and popular culture) is the totality of tradition-based creations of a cultural community, expressed by a group or individuals and recognized as reflecting the expectations of a community in so far as they reflect its cultural and social identity; its standards and values are transmitted orally, by imitation*

*or by other means. Its forms are, among others, language, literature, music, dance, games, mythology, rituals, customs, handicrafts, architecture and other arts*".

The Recommendation provided guidelines for identification, conservation, preservation, dissemination, and protection of expressions of folklore through international cooperation. States were, therefore, requested to engage in the establishment of national inventories in order to create identification and recording systems. In particular, they were requested to stimulate the creation of a standard typology of folklore by way of:

i.    a general outline of folklore for global use;
ii.   a comprehensive register of folklore; and
iii.  a regional classification of folklore, especially field-work pilot projects.

After the *World Forum on the Protection of Folklore* in Phuket, in 1997, WIPO launched nine global fact-finding missions between 1998 to 1999. The quest was to find an appropriate legal framework for regulation of folkloric expressions, which would ensure that its users achieve the objectives of a balanced IP system. In the Conclusions that accompany the *Report on Fact-Finding Missions on Intellectual Property and Traditional Knowledge* (1998–1999) we can read that "*neither existing IP standards, nor the 1982 Model Provisions, alone are sufficient in meeting the needs and expectations of indigenous and local communities, and that the testing of alternative models, using a combination of IP and non-IP measures, is desirable*" (WIPO 1999).

In 2000, WIPO decided to establish an Intergovernmental Committee on Intellectual Property and Genetic Resources, Traditional Knowledge and Folklore (IGC), aiming at creating a forum where WIPO member states could discuss intellectual property issues that arise in the context of access to genetic resources and benefit-sharing as well as the protection of traditional knowledge and traditional cultural expressions. Since then, it is noteworthy that from a terminological point of view, the terms "traditional cultural expressions" and "expressions of folklore" are now used interchangeably in WIPO discussions (WIPO 2000) https://www.wipo.int/tk/en/igc/).

Having put genetic resources and folklore under the same umbrella might be questionable as copyright law could be a better protection tool for genetic resources than for folklore, as we will discuss later.

WIPO points out that the use of IP to protect traditional knowledge and genetic resources originates from concerns expressed by indigenous communities regarding the use of new technologies, such as biotechnology, that could lead misappropriation of genetic resources and associated traditional knowledge (WIPO 2000).

IP law in this context should help to prevent and promote fair benefit sharing between holders of those assets (mostly biodiversity-rich countries) and those with the modern technologies to access and use them.

Finally, in 2009, members of WIPO agreed to develop an international legal instrument that would give traditional knowledge, genetic resources and traditional cultural expressions (folklore) effective protection. Such an instrument could range from a recommendation to WIPO members to a formal treaty that would bind countries choosing to ratify it (WIPO 2011).

In parallel to the initiatives launched by WIPO and UNESCO together, it is also to recall that in the last two decades, multiple interventions by the United Nations raised the attention to the protection of folklore and other expression of traditional knowledge from different perspectives.

In 2003, UNESCO adopted the "*Convention for the Safeguarding of the Intangible Cultural Heritage*", which entered into force in 2006 after the thirtieth ratification by Romania. As of September 2018, 178 states have ratified, approved or accepted the convention.

The 2003 UNESCO Convention, even if does not mention folklore, defines "*Intangible cultural heritage*" as "*the practices, representations, expressions, knowledge and know-how, transmitted from generation to generation within communities, created and transformed continuously by them, depending on the environment and their interaction with nature and history*". This notion of "intangible cultural heritage", includes

according to the Convention (a) oral traditions and expressions, including language as a vehicle of the intangible cultural heritage; (b) performing arts; (c) social practices, rituals and festive events; (d) knowledge and practices concerning nature and the universe; as well as (e) traditional craftsmanship (art. 2), so that folklore must be considered included.

The Convention aims at working at national as well as at international level. At national level (*III. Safeguarding of the intangible cultural heritage at the national level*), State Parties are supposed to '*take necessary measures to ensure the safeguarding of the intangible cultural heritage present in its territory*". These measures include identification of the intangible cultural heritage that exists in its territory, adoption of appropriate policies and promotion of education (art. 13).

At international level (IV. Safeguarding of the intangible cultural heritage at the international level), the 2003 Convention promotes international cooperation, which includes "the exchange of information and experience, joint initiatives, and the establishment of a mechanism of assistance" to other State Parties.

Two years later, in 2005, UNESCO adopted the *Convention on the Protection and Promotion of the Diversity of Cultural Expressions*. The convention entered into force on 18 March 2007 after ratification by 30 states.

The Convention addresses the many forms of cultural expression that result from the creativity not only of individuals, but also of groups and societies and that convey cultural content with symbolic meaning, as well as artistic and cultural values that originate from or express cultural identities (art. 4).

Among the *Measures* to promote cultural expressions, the Convention foresees at art. 7 that "*Parties shall endeavour to create in their territory an environment which encourages individuals and social groups: a) to create, produce, disseminate, distribute and have access to their own cultural expressions, paying due attention to the special circumstances and needs of women as well as various social groups, including persons belonging to minorities and indigenous peoples*".

Finally, in 2007, the United Nations adopted the *Declaration on the Rights of Indigenous Peoples* that acknowledges the right of Indigenous peoples to practice and revitalize their cultural traditions and customs. This includes the right to maintain, protect and develop the past, present and future manifestations of their cultures, such as archaeological and historical sites, but also artefacts, designs, ceremonies, technologies and visual and performing arts and literature (art. 11).

According to the UN Declaration, indigenous peoples have the right to maintain, control, protect and develop their cultural heritage, traditional knowledge and traditional cultural expressions, as well as the manifestations of their sciences, technologies and cultures, including human and genetic resources, seeds, medicines, knowledge of the properties of fauna and flora, oral traditions, literatures, designs, sports and traditional games and visual and performing arts. Even more importantly, they have the right to maintain, control, protect and develop their intellectual property over such cultural heritage, traditional knowledge, and traditional cultural expressions (art. 31).

In conclusion, we might stress that the debate on the protection of folklore at international level has been very vivacious in the last decades, although the effectiveness of the entire struggle is questionable. As it has been pointed out (Yu 2003, p. 241), the success of an international initiative in this field might depend on different issues: the forum in which the parties conduct their negotiation, the mindset itself of the negotiators, and the participation of indigenous people might play an important role in achieving effective success.

In all the cases examined in this Chapter, the communities involved in cases of cultural appropriation were rarely able to find in IP instruments an answer to their demands, which lead us to examine more in detail the concrete applicability of IP to the protection of folklore.

*4.3. The Protection of Folklore through Intellectual Property Tools*

The discourse on the applicability of intellectual property tools to the protection of folklore is full of lights and shadows.

On the one side, the numerous efforts put in place at national level as well as at international level aimed at giving local communities adequate instruments in order to protect their expressions of folklore have not always reached the goal.

On the other side, lack of protection of folklore through an IP system may end up in denying indigenous people control over their own cultural property, and indirectly on their own cultural identity (Paterson and Karjala 2003, p. 634).

Nobody doubts that as a creation of the human mind, folklore is certainly a form of intellectual property. Nonetheless, it raises particular legal and policy questions when it comes to define a specific regime of protection. In fact, the current construction of intellectual property with its eligibility criteria and limited duration is not wholly compatible with folklore protection (Asmah 2008). Munzer and Raustiala, who dedicated a long and well documented contribution to the difficulties of protecting traditional knowledge through intellectual property rights (Munzer and Raustiala 2009), point out that the debate over the legal protection of traditional knowledge is still young and new solutions should be considered.

For the time being, anyway, an overall protection of folklore under the umbrella of IP law is difficult, even if it receives a distinct focus in many national and international regimes.

It is not to be underestimated that sometimes folklore can be protected by existing legal systems, such as copyright and related rights, geographical indications, appellations of origin and trademarks (Malaurie-Vignal 2019).

If we review the different tools that IP can offer to protect folklore, we might find interesting, although limited solutions.

Contemporary adaptations of folklore, for example, are copyrightable in theory. It is not always easy to reconcile "group ownership" that characterizes indigenous communities with "individual ownership" that is one of the main features of the Western way to think about copyright. Nevertheless, although in traditional communities works tend to be created by groups, it is possible for individuals in the community to be singled out for their exceptional talent or recognized as creators (Asmah 2008). In this case, it is possible to sort out an author, who can be identified for the purposes of protection under the copyright system.

More critics come out of the fact that IP law is based on the criterion of originality, which is difficult to meet for those traditional cultural expressions that have been passed down from one generation to another.

Trademarks can also be used to identify authentic indigenous arts. Nowadays a series of successful experiences provide noteworthy examples in this field (Torsen 2010), as in the case of the the *Navajo Tribe v. Urban Outfitters*, already mentioned above. Another interesting example concerns the cooperative of *Coopa Roca*, founded in 1981 by women located in Rocinha, one of the poorest neighborhoods in Rio de Janeiro. Women working for the cooperative are artisans who use traditional handicraft techniques, such as embroidery, crochet, knitting, patchwork and fuxico—a traditional technique that involves embroidering with pieces of fabric (Torsen 2010). The cooperative has registered its own trademark Coopa Roca and makes now expensive high-fashion clothes sold all over the world (Torsen 2010).

Notwithstanding the success of many commercial initiatives (Torsen 2010), it is noteworthy to recall that trademark protection extends only to uses in commerce, but might be inadequate to protect folklore outside the commercial use (Vézina 2019).

Geographical indications and appellations of origin may also play a role in the protection of folklore (Malaurie-Vignal 2019; Torsen 2010). A geographical indication (GI) is a name or sign used on certain products, which corresponds to a specific geographical location or origin. The use of a GI acts as a certification that the product possesses certain qualities, or enjoys a certain reputation, due to its geographical origin. A geographical indication right enables those who have the right to use the indication to prevent its use by a third party whose product does not conform to the applicable standards.

Appellations of origin are a special kind of geographical indication. GIs and appellations of origin require a qualitative link between the product they refer to and its place of origin. Both inform consumers about a product's geographical origin and the characteristics of a product linked to its place of origin. The basic difference between the two concepts is that the link with the place of origin must be stronger in the case of an appellation of origin. The quality or characteristics of a product protected as an appellation of origin must result exclusively or essentially from its geographical origin. However, a protected geographical indication does not enable the holder to prevent someone from making a product using the same techniques as those set out in the standards for that indication (WIPO, *Frequently Asked Questions: Geographical Indications*, http://www.wipo.int/geo_indications/en/faq_geographicalindications.htm). Products identified by a geographical indication are often the result of traditional processes and knowledge carried forward by a community in a particular region from generation to generation (Malaurie-Vignal 2019). Similarly, some products identified by a geographical indication may embody characteristic elements of the traditional artistic heritage developed in a given region. This is particularly true for tangible products such as handicrafts, made using natural resources and having qualities derived from their geographical origin.

GIs do not directly protect the subject matter generally associated with folklore, which remains in the public domain under conventional IP systems. However, GIs may be used to indirectly contribute to their protection, for instance, by preserving them for future generations. This can be done, for example, through the description of the production standards for a GI product, which may include a description of traditional processes or knowledge (Malaurie-Vignal 2019).

One interesting example based on geographical indication concerns the *Kullu Shawls* in India (Torsen 2010). Kullu is located in a valley in the mountains of Himachal Pradesh, in the northern part of India, in Western Himalayas. Kullu shawls contribute significantly to the economy of the valley, where most of the inhabitants earn their living by weaving part time or full time. The Kullu shawls are part of local folklore. They are woven using handlooms that can be found in almost every home, following a traditional knowhow that local people have inherited from their ancestors.

In this context, the Kullu Shawls Weavers Association (KSWA), took the initiative and registered Kullu Shawl as a geographical indicator in 2006 (Simha 2009). According to the *Geographical Indication of Goods Act of 1999*, the GI, if correctly implemented, prevents sale of non Kullu shawls. It also prevents proprietors from using the name "Kullu shawls" if they are produced outside the defined geographical territory of Kullu Valley Region of Himachal State. Further, unauthorized shopkeepers or producers cannot even use signs boards/hoardings of selling Kullu shawls. In the event of anyone found selling fake shawls, a huge penalty will be imposed on them or imprisonment of 6 months to 3 years or both under the GI Act of 1999 (Simha 2009).

Despite the recognition as GI, local observers point out the difficulties in implementing the law. The Kullu Shawls Weavers Association encountered many difficulties in identifying the number of local household and commercial weavers whose shawls could be given a GI mark to protect them from imitations. The process has been slow because the weaving industry in the Kullu valley is not organized, as weavers are scattered in remote parts of the valley.

In conclusion, for local communities it is difficult and sometimes frustrating to overcome the difficulties that arise in establishing elements of traditional intellectual property protection (Paterson and Karjala 2003, p. 634; Farley 1997, p. 1). Even in the cases where the particular country has a formal law that protects folklore, it is hard to ensure compliance overseas for copyright infringement related to folklore (Asmah 2008, p. 6). That is why many voices have begun to stress the possibility of looking beyond intellectual property in protecting folklore.

## 5. Looking for New Solutions to Protect Folklore in the Fashion Field

*5.1. The Actual Needs behind the Protection of Folklore*

In the search for new solutions, it is important to investigate the actual needs of those peoples who vindicate their rights about their own folklore.

From a first inquiry, what comes out is not just the request of a static protection of the given expressions of folklore, but more a need of collaboration in order to render folklore a way to enhance the economy of local communities, while respecting their identities and values. According to recent researches, folklore cannot only be understood as a source of identity, but also as a resource for sustainable development (Coombe and Turcotte 2012). Many initiatives have led to a global awakening of the importance of artisans and their culture to a sustainable world. For example, according to the 2011 *Crafts Economics and Impact Study* (CEIS) by the *Crafts Council* of India, *'there is recognition of the living fabric of community and social relationships that go beyond monetary value.'* Increased environmental awareness has added significance to the fact that artisans prefer to use natural resources in the creation of their products (Emmett 2014).

We might say that in the protection of folklore the debate is characterized by two contrasting approaches, one that tends towards preservation of folklore in its original form and the other that connects the survival of folklore to innovation. In this perspective, the first approach tries to apply intellectual property rights to folklore in order to protect its authentic and original form. The second approach, on the contrary, tends to look to folklore in a dynamic perspective that sees in innovation the only way to sustain traditional culture on the long run (Pager 2012).

In this perspective, legal provisions on cultural property do not always seem the best solution to enhance folklore, as they allow the "*assertion of fixed moral and economic ownership over fluid cultural practices and identities*" (Aragon 2012). Legal rules sometimes seem to harness realities that are in constant motion and evolving. Textiles, that are now described as "traditional", are often the result of a long and constant evolution (Meneses Lozano 2014).

Analyzing some practical examples that are described by non-Western Authors (Prempree et al. 2014; Fukatsu 2014) the needs that arise do not seem to coincide with the attempt to protect folklore in its current form, as if to crystallize it for posterity.

On the contrary, the request seems much more directed to receive collaboration in order to ensure that original products, although remaining anchored to identity values of a certain culture, can evolve to meet the tastes of today's society.

This operation, which confronts artisans with the opportunities, but also with the challenges of new markets, is not without risks. If it is true that many of the craft-based products are by their very nature too traditional to appeal to a broad global market, it is also true that by changing the aesthetic of the original product to a more westernized aesthetic, the connection to the cultural history of the artefact may be lost. Additionally, it has been suggested that artisans may be persuaded to make products with which they may have little emotional connection (Bissett-Johnson and Moorhead 2018).

In a recent analysis of the silk fabric in Thailand (Prempree et al. 2014), the Authors were pointing very clearly out that "*Silk fabric in Thailand is a creative artwork that must be developed and modernized to adapt to the requirements of modern markets in order to enhance the economy*" (Prempree et al. 2014, p. 305). While emphasizing the identity value of silk processing, the inquiry pointed out that: "*for the most part, woven silk products are made in the original form and there has been no development or adaptation to modern society*" (Prempree et al. 2014, p. 305).

The idea from which the Authors start is that in order to add value and to let silk products flourish, traditional knowledge must be integrated with new tools that come out of a creative economy approach that would render traditional products more appealing in today's competitive environment. In this perspective, we find a request to protect folklore through innovation.

The investigation on silk fabric in Thailand focuses on some critical aspects that should be innovated in order to maintain folklore alive, not only as an identity value, but also as a way to enhance community economy.

Among these critical aspects, the Authors point out that:

- the equipment used in the production process has not been modernized;
- the production is uncertain because the weavers lack knowledge of systematic management;
- the product sale prices are not standardized because the weavers do not know how to allocate prices;
- the product design does not meet the needs of the market because there is no information or research about consumer trends and preferences;
- the marketing of products is practically non-existent due to a lack of education;
- if there is high demand the weavers are unable to produce goods in time to meet it;
- there is no quality control process.

If we look into this variety of factors, a consideration arises spontaneously: no request arises to maintain folklore as it is, but—on the contrarya struggle appears evident in the way antique ways of producing beautiful artifacts can be rendered more adaptable to today's needs. This aim will not be achieved only through IP protection, but it could be made possible through collaboration with the holders of certain skills that generally work for Western designers.

Another example in this line is provided by an interesting project launched in Japan in order to revisit traditional textile designs in a more sustainable way (Fukatsu 2014). Here, the idea is to point out that "*production of traditional textiles by local communities often tends to become too traditional and old-fashioned separating from our lives and social life*" (Fukatsu 2014, p. 2).

In Japan, the current situation aims at protecting people who are holding traditional techniques in a static way, as cultural heritage and museum collection. The project suggests a different approach, reexamining traditional textiles in order to design new ideas and products for social innovation toward sustainability. One major example concerns the use of banana fibers that are generally thrown away after harvest, as material to be used for the production of environmentally friendly products. In this way, waste materials can be reused as valuable materials.

Using traditional techniques, it was possible to develop a method to extract fibers from fruit bananas, to make yarns, fabrics and papers. As in Japan there are no banana plantations and almost all fruit bananas have been imported from tropical areas, the purpose of the project is to cooperate with foreign countries where bananas are produced (Fukatsu 2014, p. 3).

These various examples lead us to the topical question: "*Is it possible to create productive collaboration across cultures without exhausting or dispossessing the custodians of tradition?*" (Ballenger and Hamlin 2018).

*5.2. Towards Possible New Solutions*

Looking for possible new solutions in the field of cultural appropriation in the fashion world, it is important to bring to the attention that besides legislative initiatives, an important role may be played by *private governance* tools and, in particular by *Corporate Social Responsibility* standards. *Private governance* can be defined as the phenomenon of private actors pursuing public goals and interests while exercising traditional state functions in the forms of rulemaking, implementation and dispute resolution.

In recent years, we have witnessed an increasing interest amongst private actors in voluntarily pursuing public values and interests such as human rights, social policy, health and environmental protection through private standard setting (Pattberg 2005; Falkner 2003; Knill and Lehmkuhl 2002).

In the fashion industry, compliance with *Corporate Social Responsibility* principles is developing rapidly and almost all major fashion brands have links to their corporate code of ethics or code of conduct (Cerchia and Piccolo 2019). Many contemporary designers follow social standards in their

design practice, some through their work with traditional textile artisans including artisan techniques in their designs (Emmett 2014).

In some challenging fields, the trade associations that represent the fashion world have launched specific voluntary initiatives that aim at fixing behavioral standards for business, while enhancing awareness among consumers. In 2012, for example, Camera Nazionale della Moda Italiana together with Confindustria Moda, have launched a "*Manifesto for the sustainability in Italian Fashion*" that interprets the global challenges of sustainability by defining concrete and distinctive actions to be taken by Italian businesses. The Manifesto pursue various goals: it shows to Italian companies how to take advantage of the opportunities offered by the greater attention given to environmental and social aspects and, at the same time, it helps them to manage reputational and operational risks in the best way.

The same approach should be followed as far as cultural appropriation is concerned, in order to extend the principles developed in the field of environmental sustainability to social sustainability.

Fashion industries associations should become the interpreters of a new way of approaching the problem of cultural appropriation, establishing self-regulation mechanisms, like standards accompanied with a certification label (Vézina 2019).

Collaboration between Western designer and local artisans should be encouraged, in order to foster sensitivity and understanding between artist, designer, and eventually consumer. Cultural sensitivity, education, and ethical business practices will teach values and encourage conscious choices for industry and also for consumers (Ballenger and Hamlin 2018).

In the wake of the initiatives launched by UNESCO, private business should take advantage of the several results already attained.

For example, Craft Revival Trust together with UNESCO has released a study ("*Designers meet Artisans*", 2005), that outlines several projects that have already reached important achievements where designers and artisans have developed or co-designed new products for new markets. This publication suggests several different models for designer-artisan collaborations in both India and South America, concluding with a list of guidelines concerning the different ways to undertake collaborations.

UNESCO has also launched a Traditional Knowledge World Bank (TKWB) Web system platform designed for the dissemination and sharing of Traditional Knowledge while simultaneously paying respect to its origins (http://www.tkwb.org/w/index.php?title=TKWB). In the framework of its activities a specific "Creative Textile" project has been launched, whose main aim is to preserve, narrate and share textile traditions and their ancient techniques in order to create a cross-pollination project among textile artisans of different background and nationality (https://www.itkius.org/creative-textile-2/). The project aims at documenting and preserving ancient traditions and knowledges. The project further tries to engage communities and territories known for their traditional textile techniques into a common purpose, that is to create a stronger relation among communities around the world.

Many examples show that the idea of becoming more responsible towards local communities is spreading also in the industrial sector. Oskar Metsavaht has launched Osklen, a brand that gives priority to those productions that can help to emancipate the communities involved, creating new sources of income and giving them the tools to collaborate not only with Osklen, but also with other brands. 'As Sustainable As Possible, As Soon As Possible' philosophy, is constantly evolving ways to enhance the sustainability scope of the brand, supported by Instituto-e, a non-profit organisation dedicated to sustainable development.

Metsavaht, is also the founder of *Instituto-e*, a Civil Society Organization of Public Interest, that proposed the ASAP project: *As Sustainable As Possible, As Soon As Possible* (https://osklen.com/pages/ethics-and-sustainability)

The institute further develops and implements environmental projects inspired by the concept of the *6 E's*: Earth, Environment, Energy, Education, Empowerment, and Economics.

The projects developed until now concern training, empowerment and income generation to small communities and groups of women that live in vulnerable conditions inside Brazil and abroad.

## 6. Conclusions

Cultural appropriation is a complex phenomenon. Lawyers who try to approach the issue should contextualize the problem from a historical perspective, trying to apply the necessary tools to cope with all the different facets that the phenomenon presents.

Indigenous people should have the possibility to access conventional intellectual property claims whenever these can allow a fair compensation for cultural appropriation.

Nevertheless, intellectual Property is not always the answer. The needs of indigenous people have to be understood and supported, taking into account their own perspective that might not coincide with Western lawyers' before imposing new laws that are oriented to defining property for profit over creative practices that are oriented to multiple social values and local livelihood goals (Aragon 2012).

That is why, next to IP tools, other solutions should be envisaged that allow folklore not only to be protected, but also to develop according to that natural and ancestral flow, which passes from generation to generation, helping to build an important identity, but at the same time guaranteeing the possibility of adapting to the survival needs of that culture.

**Funding:** This research received no external funding.

**Conflicts of Interest:** The authors declare no conflict of interest.

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

Zografos, Daphne. 2010. *Intellectual Property and Traditional Cultural Expressions*. London: Edward Elgar Publishing, (The States that enacted specific statutes in order to protect their folklore are: Algeria in 1973; Senegal in 1973; Kenya in 1975; Mali in 1977; Burundi in 1978; the Ivory Coast in 1978; Guinea in 1980; Cameroon in 1982; Congo in 1982; Madagascar in 1982; Rwanda in 1983; Benin in 1984; Burkina Faso in 1984; the Central African Republic in 1985; Ghana in 1985; Zaire, 1986; Nigeria, 1988 and 1992; Lesotho, 1989; Malawi, 1989; Angola, 1990, Togo, 1991; Niger, 1993. In Central, South America and the Caribbeans: Bolivia, 1968 and 1992; Chile, 1970, Colombia, 1982; Barbados, 1982; Dominican Republic, 1986; Panama, 1994. In Asia Iran, 1970; Sri Lanka, 1979; Indonesia, 1987).

