# Peer review of "Fashion between Inspiration and Appropriation"

_laws_

Round 1
Reviewer 1 Report
This is a generally interesting article with some merit, but there are problems with the way it has been written and general issues with language and structure. The very short punctuated paragraphs make it a difficult read to start with. The focus on folklore is also quite baffling. I think they mean 'traditional cultural expressions' which is the WIPO term for expressions and designs which are broader than folklore. Usually folklore is just used for stories. I think they need to rethink the way they've written that side of things.
With some revisions to this and to the paragraphs to enable greater flow, it would read better. Also, the paper could more clearly explain if there was a methodology used here. It seems mostly literature review/review of laws. Many examples are given, but most are given with minimal depth, and perhaps not enough critique. More could be done to improve this side of things.
Author Response
Thank you for the suggestions.
I have changed the structure with short punctuated paragraphs that rendered my article a difficult read the whole article has been revised by a native speaker As far as the choice of using the term “folklore” concerns, I discuss my choice at page 8, where I point out that "expressions of folklore" are included in the definition of Traditional Cultural Expression (TCE) by WIPO and may refer not only to stories, but also to “music, dance, art, designs, names, signs and symbols, performances, ceremonies, architectural forms, handicrafts and narratives, or many other artistic or cultural expressions.Reviewer 2 Report
Overall
The topic (cultural appropriation in fashion) is a timely topic (insofar as it has received media coverage recently) but one which has been covered in specialist literatures to a degree. As a scholarly discipline, the law has paid some attention to this question but there is much more to be added, and the authors could well be the ones making that contribution.
However, the submission bears a number of flaws or caveats which makes the analysis fall short of making a valuable and original contribution to the legal scholarship on this. These flaws include the fact that: the area(s) of law consulted are not clearly outlined or explained. The authors discuss TCEs protection, IP rights, and international law such as the UNESCO convention without a clear explanation of how these rights might work together, overlap or apply alongside one another. The discussion also fails to explain what cultural appropriation in fashion might look like in practice, and how these rights might help address the issues outlined (or not). Anecdotes are gives, and cultural appropriation are discussed in general or abstract terms, but these developments are too far remove from practice to give a concrete sense of how the law (international or national) might help or hinder cultural appropriation or creativity in fashion.
Comment on the content:
The authors need to define what they mean by 'West' (and 'Western') and 'East' (and 'Eastern'). They run the risk of coming across as broad-brushed, generic or too general in their analysis otherwise. It is also important to define key terms used in the paper such as 'Orientalism' and 'Occidentalism' at the start of the paper/analysis. This is especially important here as the audience will likely be a readership trained in law, not cultural studies.
The authors also need to clearly explain (in the introduction): the structure of the paper, which literature or body of knowledge they will use to enrich their analysis of the law, and very importantly which area of law they are consulting in their discussion of cultural appropriation in fashion- and why. These clarifications are critical but are currently missing from the piece.
All quotes need to be properly referenced and sourced, including the opening one (Cf. Coco Chanel).
The choice of jurisdictions needs to be explained.
Minor stylistic points:
- the paper has already heavily relied on direct quotes from other authors. It is makes for a slightly harder read because the flow of the piece is interrupted by the writing and style of others. It would be good to integrate the contribution of the literature within your argument, without as much direct quotations as it detracts from your main argument(s)
- I would avoid one-sentence paragraphs (eg: lines 112-114)
Author Response
I have added a new introduction where I explain the structure of the article The concept of East and West are clearly discussed The quotes have been added One sentence paragraph have been avoided The whole paper has been revised by a native speaker I prefer to mantain the quotesReviewer 3 Report
I want to start by saying that I enjoyed reading it. I found the first eight pages fascinating. I had never read a law article that analyzed cultural heritage and appropriation using fashion. I used enjoyed the broader insights into fashion as an element of culture and the other functions of fashion besides as an indicator of socio-economic class. Now, I think that it is clear that I am probably not the best peer/expert reviewer for the first eight pages. However, the statements and conclusions are consistent with the other areas that I am more familiar with the appropriation of art, crafts, or music (broadly folklore). I will defer to other reviewers on these topics.
If I were to identify two weaknesses, they would be; first, I was not sure about the thesis or purpose of the paper until I was well into reading it. I would think about stating the thesis or objectives of the article in the first few paragraphs would situate the reader to understand the first eight pages better. Second, the fashion motif is fascinating. But, it only carries the first eight or so pages then disappears on page 8. I think the article would be stronger and make a more significant contribution if that motif carried throughout the piece. I also have a curious question, do you think that folklore is limited to "indigenous peoples." After reading your article, folklore could also protect the jazz music of New Orleans (USA) or other marginalized peoples who create culture artifacts on a collective individually unidentifiable level. Did you intend to include these groups? I just think that's an interesting idea.
These are a few examples of my minor quibbles, comments, and questions.
Line 20, a citation as to source, would be helpful. The source I found was Life Magazine, 19 Aug 57—there may be better sources.
Line 33, I find the capitalized use of the word author a bit confusing, I am not sure if you are referring to the author of the quote or reference or the Author of the paper. It may be better to use the names of the author(s) of the materials rather than the word author. This is a style comment, and it's a matter for the editors.
Line 42, the author of the article, is well aware of the problematic nature of the term Orient. In one reading, it seemed very appropriate in the historical-cultural context, and in a subsequent reading a bit jarring, I merely call this to the author's attention.
Line 44, dating back 20 years, I think this means the book is 20 years old rather than the book covers the past 20 years. For me, this was a bit confusing. I immediately thought of the European fashion influences in Shanghai in the 1920s and 30s.
Line 333, under US law, a cease and desist letter is sent by an attorney requesting that some action stop and explaining why the act is prohibited, it is not "issued." A court may eventually is some proceeding issued an injunction. I cannot find an order in the opinion. See https://narf.org/nill/bulletins/federal/documents/navajo_nation_v_urban_outfitters_2016.html. The judicial opinion seems to focus on the validity of NAVAJO as a trademark. I think that the author may not fully appreciate the use "in commerce." It is a term of art in US law and does not mean what it says. "In commerce' means all commerce that the US Congress can lawfully regulate under the Commerce Clause of the US Constitution. Art. I, sec. 8, cl. 3 "To regulate Commerce with foreign Nations, and among the several States, and with the Indian Tribes." This is the constitutional basis for federal trademark registration. Under US law, the US of any instrumentality of interstate commerce, for example, an interstate highway may be sufficient or anything that affects interstate commerce, There may be more room here for legislation that the author acknowledged. Also, individual states could always pass laws protecting cultural heritage as long as state law was not federally preempted.
Anyway, this is more than the author or editors want to know about the commerce clause.
Line 363, I don’t think the word arguing is the best word to use here, holding, concluding
or finding would be more suitable terms. On line 364, I would change couldn't to could not.
Line 383, I question the use of native cultures. This may be the best word to express the author's meaning, but the word has many connotations in English.
Line 477-503, I would like to know a bit more about most of these efforts. They are just listed without any real detail or analysis. For example, lines 490-492, The 1967 Stockholm Conference on the Berne Convention what happened, what was the issue the "entered the agenda" and perhaps why it did not proceed? The author raised my curiosity with the litany of sources and historical events but left me unsatisfied. Developing these topics would be especially useful to individuals like myself who may not be familiar with all of the attempts at protecting "folklore" in Africa. I noticed that there is no discussion or even listing of Latin American efforts to protect folklore. For example, Brazil, in 1937 and Mexico in 1946, passed laws to defend their cultural patrimony. Of course, either the geographic region or these laws may be outside the scope of this article as these early laws tended to focus on physical artifacts. If there is nothing relevant in these regions, this should be stated.
Lines 583-585 How successful were these model laws? It would be helpful to given a concise evaluation or opinion if this is available.
Line 545 think about changing doesn't to does not.
IP & Folkore—there is a huge body of literature on this topic. I think articles such as the Kuruk article cited in this article develop this very well the challenges that folklore poses to the traditional IP regime. The article would be much stronger if the author delved more deeply into these issues. Issues of authorship (ownership) and issues of control must be resolved. For example, in some of the laws, the license to use is given by the nation-state and not by the indigenous creator-owners of the folklore. Thus, there is no guarantee that traditional values would be respected. It is identifying who or what in the community controls and can authorize its use that is critical to its protection and ensuring the benefit inures to the appropriate community. What is the community? Often folklore creators are not contiguous with national boundaries. How big a group or how long must something exist before it achieves this status? The duration as noted by the author is also a problem. Folklore should be protected as long as the culture survives.
Lines 635-640 The Berne Convention except in one provision presupposes an identifiable human author(s) and not a collective unidentified author. Also differences, for example, France protects fashion under its copyright laws while the USA does not. I am not sure the Berne is either the floor or ceiling in this matter. What about unfair competition law?
Lines 651-653, what about certification marks?
Lines 656-passim. I would develop GIs more since the ownership of a GI tends to be controlled by members of the GI product-producing group or a political body accountable to them. Also, the group may define the GI . This collective, potentially long-duration protection may be enough. Right now, GIs tend to be agricultural products, but there are moves to extend the scope of GIs to goods manufactured traditionally. I would develop this area a bit more.
Lines 741 why is this subversion bad? I am thinking about the China porcelain export trade if the 19th Century. It supported local artists who could then afford to meet local demands.
Lines 749-786 I wonder though if there is an active market for "export" or non-indigenous consumption of folklore will the monetary market dry up the production of tradition products for traditional use. This market approach seems to guarantee that the skills will remain but not the actual products in their traditional forms. I am reminded of seeing traditional woven patterns done in florescent colors when I was in Kunming.
Lines 791-795 I suggest expanding this corporate social responsibly discussion. The ability of Ethiopia and I think Jamaica to recapture trademarks named after local regions from Starbucks, for example. This market-consumer pressure may work. I think that all of the examples given in the first part of the paper market forces or social pressure was more effective that law in resolving the rights issues.
Line 853-862 There is no discussion of sui generis methods of protecting folklore. The author suggests that there should be more or in other ways. The author should consider adding a brief discussion of sui generis protection, whether it has actually been implemented or merely proposed. For example, Could the French fashion design law (part of French IP law) be useful here to protect traditional designs? I have no idea I'm just speculating. I think that over the years, WIPO has published a series of reports or documents on ways of protecting traditional culture. Some of these may have interesting ideas as to sui generis protection.
Author Response
I really would like to thank Reviewer 3, because it is the only one who really helped me improving the paper (at least I hope)
I introduced all the detailed corrections indicated in the review: ine 20, line 33, line 42, line 44, line 333, line 363 The article has been revised by a native speaker I have added a new introduction in order to better explain the sructure of the paper I did not add all the parts required by the Reviewer because they are already covered by other articles of the special issue on "The new frontiers of fashion law"Reviewer 4 Report
The article is interesting and deals with a topic on which legal letterature is quite abasent.
The large numeber of examples drawn from a global point of view enriches the value of the essays.
Author Response
The Reviewer was already satisfied by the paper
Round 2
Reviewer 2 Report
The structure of the paper and the introduction is much improved. The following points (some more important than others) would need to be address in order for the article to reach its full potential.
The authors have clearly done a lot of work assembling references and examples of appropriation. There is also a lot of work done on mapping relevant literatures (yes, plural!) across different field. These are the strengths of the paper. However, more needs to be done to integrate all of this information into a more coherent, and tighter, argument. The argument of the authors, and their original contribution to the debate of cultural appropriation in fashion is not clear. if the aim of the paper is to deliver a complex literature review (and newspaper review), which could be a valuable exercise indeed, this is absolutely fine, but this must be explained in the introduction.
At the moment the paper has two heads:one more theoretical discussing the tensions between Global North designers using or appropriating cultural heritage from other cultures or countries. The other head turns to the law. This second part is the least developed of the paper, and needs more attention. It is particularly important to present the various frameworks applicable to cultural heritage in a more synthetic manner. Avoid listing or cataloguing initiative and present the state of the law as a whole. It is true that international and national laws are very fragmented. Better state it that way and point to examples than list separate instances of policy-making or reforms. This will help you cut down on words and be clearer in your argument - in my view.It is not clear what is the original contribution of your paper with regard to the law, despite the fact that the introduction (and the paper) signals that to be the point of your analysis.In particular the last part on 'towards new solutions' does not detail the content of these new proposals (the Manifesto, and the Report) and therefore does not explain how these proposals differ from existing policies or frameworks. That part of the paper is very important and is too 'thin' at the moment. It needs to be unpacked further to deliver on the promises you make in introduction. To give space to this part of your paper, I would cut down on Part 2 (Look into the past).
Aside from this general and more important points, I would consider addressing the following points in your next/new draft:
References to support the first sentence of your introduction would be welcome (l 69 to 70) Line 74, I would replce by "the paper" to avoid confusion. Throughout the whole piece, I would avoid one-sentence paragraphs as a matter of style. (see e.g. line 92, line 263, line 554). One-sentence paragraph can often be integrated into the preceding or the following paragraphs (or sometimes deleted). This might help you shave off precious words/sentences to "spend" on other parts of your reasoning. Section 2 of your paper, and the first half of the paper, is very reliant on quotes. It is not always clear why those quotes are needed? I would encorage paraphrasing and referencing these instead of disrupting the flow of your writing with someone else's words. An example of quotes which perhaps is not needed can be found on line 123-127 (retired French missionary) Consider deleting Section 2 entirely, to develop and hone your section on the law. This would be a better fit for the readership of the journal, and for the scope of the paper as you outline it in introduction The definitions of East and West under Section 2 (look into the past) should come up in the introduction. The introduction is much improved in this version of the manuscript, mainly because it now gives a clear path for the argument. However, it would be good place to delineate some of your key concepts, so that it is clear in the head of the reader what you will talk about and what concepts will shape your reasoning. Concepts I have picked up on which seem central to your arguments are: inspiration vs appropriation vs commidification, Fashion vs folklore vs traditional cultural expression (TCE), intellectual property (making a different there between sui generis rights in cultural heritage, copyright and design rights), cultural heritage regulation. If you do that in the introduction you will find that other sections will be streamlined. Overall the paper is too reliant on quotes. Some para of your paper are almost entirely made up of quotes for no obvious reasons. Again I would remove these, and paraphrase in your own words. Examples of these para are: lines 274-279; 280-284; 292; 536 Can you clarify whether you think inspiration is a form of appropriation? At present one para on inspiration is followed by a para on inappropriate appropriation - do you mean that the examples of inspiration you gave could fall in that second category? Should website links appear as in-text references? Shouldnt the author's name or organisation be referenced instead? (e.g. line 501) The title of 3.4. should be changed, it is too vague at present. 'Making a point about appropriation in fashion' is the purpose of the whole paper, so it cannot be attributed to a single section (not logically). Sentence at lines 525-527 does not seem to fit the new theme introduced in this section - is it the best place for it? Consider deleting.